# Improving tuberculosis case detection through contact risk stratification by Xpert MTB/RIF Ultra and spatial parameters: Evaluation of an innovative active case finding strategy in Mozambique (Xpatial-TB)

Belén Saavedra[1,2], Dinis Nguenha[1], Laura de la Torre-Pérez[1,2], Edson Mambuque[1], Gustavo Tembe[1], Laura Oliveras[3,4], Matthew Rudd[5], Paulo Philimone[1], Benedita Jose[6], Juan Ignacio Garcia[7], Neide Gomes[1], Shilzia Munguambe[1], Helio Chiconela[1,6], Milton Nhanommbe[1], Santiago Izco[8], Sozinho Acacio[1], Alberto L. García-Basteiro[1,2]*

1 Centro de Investigação em Saúde de Manhiça (CISM), Maputo, Mozambique, 2 ISGlobal, Hospital Clínic —Universitat de Barcelona, Barcelona, Spain, 3 Agència de Salut Pública de Barcelona, Barcelona, Catalonia, Spain, 4 Institut d'Investigació Biomèdica Sant Pau (IIB Sant Pau) Barcelona, Catalonia, Spain, 5 Department of Mathematics and Computer Science, The University of the South, Sewanee, Tennessee, United States of America, 6 National Tuberculosis Control Programme, Maputo, Mozambique, 7 Population Health Program Texas Biomedical Research Institute, San Antonio, Texas, United States of America, 8 STD/HIV/Aids and Tuberculosis department, Ministry of Health and Social Welfare, Malabo, Equatorial Guinea

* alberto.garcia-basteiro@manhica.net

## Abstract

Prompt diagnosis is critical for tuberculosis (TB) control, as it enables early treatment which in turn, reduces transmission and improves treatment outcomes. We investigated the impact on TB diagnosis of introducing Xpert Ultra as the frontline diagnostic test, combined with an innovative active-case finding (ACF) strategy (based on Xpert Ultra semi-quantitative results and spatial parameters), in a semi-rural district of Southern Mozambique. From January-December 2018 we recruited incident TB-cases (index cases, ICs) and their household contacts (HCs). Recruitment of close community contacts (CCs) depended on IC's Xpert Ultra results, and the population density of their area. TB-contacts, either symptomatic or people living with HIV, were asked to provide a spot sputum for lab-testing. Trends on TB case notification were compared to the previous years and to those of two districts in the south of the Maputo province (control area), using an interrupted time series analysis with and without control (CITS/ITS). A total of 1010 TB ICs (37.1% laboratory-confirmed) were recruited; 3165 HCs and 4730 CCs were screened for TB. Eighty-nine additional TB cases were identified through the ACF intervention (52.8% laboratory-confirmed). The intervention increased by 8.2% all forms of TB cases detected in 2018. Xpert Ultra *trace* positive results accounted for a high proportion of laboratory confirmations in the ACF cohort (51.1% vs 13.7% of those passively diagnosed). The Number Needed to Screen to find a TB case differed widely among HCs (55) and CCs (153). During the intervention period, a reversal of the previous negative trend in lab-confirmed case notifications was observed in the district.

**Data Availability Statement:** A Supporting Information file has been attached with the study dataset, and the registries to replicate the ITS&CITS analysis.

**Funding:** This study was funded by the Stop-TB Partnership, as part of the TB REACH programme: Supporting Innovation in Detection and Care for Tuberculosis (wave 5). We acknowledge support from the Spanish Ministry of Science, Innovation and Universities through the 'Centro de Excelencia Severo Ochoa 2019–2023'. Programme (CEX2018-000806-S),and support from the Generalitat de Catalunya through the CERCA Programme (B.S, L.T.P and A.G.B). B.S received a pre-doctoral fellowship from the Secretariat of Universities and Research, Ministry of Enterprise and Knowledge of the Government of Catalonia and co-funded by European Social Fund (AGAUR). The funders had no role in study design, data collection and analysis, decision to publish, or preparation of the manuscript.

**Competing interests:** The authors have declared that no competing interests exist.

However, the CITS model did not show any statistically significant difference compared to the control area. Paediatric population benefited the most from the ACF strategy and HCs screening seemed an effective intervention to find microbiological confirmed cases in early stages of the disease.

## Introduction

Tuberculosis (TB) remains one of the leading causes of death at global scale. In 2022, it was responsible for 10.6 million episodes of disease and 1.3 million deaths [1]. Prompt diagnosis is critical for TB control, as it enables for early treatment, which reduces transmission and improves treatment outcomes due to diagnosis in the early stages of the disease [2]. However, underdiagnosis and underreporting are major challenges for surveillance systems [3], and large gaps between notifications and estimates of incidence are shown in several high burden countries. Overall, it has been estimated that over 4 out of 10 TB cases were not diagnosed and/or reported to the health authorities in 2022 [1].

TB diagnosis under National TB Programs (NTP) relies predominantly on passive case-finding strategies (PCF). They require individuals to recognize their symptoms and attend healthcare centers to be diagnosed by health care workers. Conversely, active case-finding (ACF) activities allow the detection of asymptomatic or symptomatic non-healthcare-seeker cases through systematic screening and evaluation of populations at high risk of developing TB [4]. ACF strategies constitute a useful attempt to increase detection and reduce diagnostic and reporting gaps. In addition, individual health care and counselling might modify risk perceptions and healthcare-seeking behaviors [5]. However, ACF interventions are extremely diverse and often tailored to different population characteristics and resources [6, 7]. In fact, reported effectiveness of ACF activities are inconsistent and their impact on TB control and patient outcomes needs to be rigorously evaluated [8].

Mozambique is one of the countries with highest burden of TB, multidrug-resistant TB (MDR-TB), and TB/HIV. In 2017, the estimated detection gap was 48% [9]. While recent data show an improvement in this estimates, there are some concerns about the quality of diagnostic and reporting services, given the worrying low bacteriological confirmation rate among reported cases [10]. Several factors could contribute to this low case notification, including structural limitations of the health system, poor perception of health risk, or traditional health beliefs among others [11–13]. Besides, diagnosis in people living with HIV (PLHIV) is challenging [14], and hinders TB control in settings where both epidemics co-exist. Thus, the implementation of ACF interventions in high-risk populations could contribute to reduce the burden of missing cases [15].

Household members and people who interact frequently with a TB case (household contacts, HCs or close community contacts, CCs) are at higher risk of TB than the general population, due to the longer time of exposure [16]. As the bacillary load of index cases (IC) could be a proxy indicator of infectiousness [17], this could be used to stratify transmission risk. Thus, we designed an innovative ACF strategy, based on the combination of Xpert MTB/RIF Ultra (Cepheid, Sunnyvale, CA, USA, hereinafter Xpert Ultra) results and spatial parameters on notified tuberculosis cases (overall and microbiologically confirmed) in a high TB and HIV burden district of Southern Mozambique. The main objective of this analysis is to assess the impact of this combined intervention on TB case detection.

## Methods

### Study design, population and setting

Over a period of 12 months (January-December 2018) the whole district of Manhiça participated in the Xpatial-TB project (TB REACH, wave 5, funded by the Stop TB partnership). This was an intervention study which introduced both: i) an ACF initiative, and ii) the novel Xpert Ultra assay as the frontline test for active TB diagnosis (as it has been described elsewhere [18]). Manhiça (intervention area) is a semi-rural area, 80 kilometers away from the capital, with a population of approximately 205,000 inhabitants as of 2019 [19]. The study was implemented by the Manhiça Health Research Center (CISM, from its acronym in Portuguese), which runs a Health and Demographic Surveillance System (HDSS) since 1996. This HDSS routinely collects sociodemographic information and actively follows important demographic events of nearly 100% of the population in the district [20].

### Xpatial-TB strategy

During the study period, new and relapse TB patients (clinically diagnosed or laboratory confirmed) who started treatment in Manhiça district were invited to participate in this study. Those providing informed consent were included as index cases (IC) of the strategy and followed the study algorithm shown in Fig 1 (Additional information in S1 Text Xpatial-TB procedures details).

**ACF activities.** Once an IC was identified, a list of households (HCs) and close community contacts (CCs, eligible neighboring residents) was generated based on HDSS registers.

The contact population identified per IC depended on two main variables: the bacillary burden of the IC, and the household density of the neighborhood where the IC lived in. The bacillary burden was classified as per Xpert Ultra semiquantitative results (*High/Medium/Low/Very low/Trace/Negative*). Following our algorithm, ICs with *low, medium or high* Xpert Ultra results, generated both HC and CCs. Those ICs with a *very low, trace or negative* result, only generated HCs (based on studies relating infectiveness and bacillary burden [21]). The area around which neighboring contacts were screened depended on predefined radii from the ICs' household. These radii were established based on the household density of the neighborhood where the IC lived in.

After ICs' enrolment, three field teams, composed of one nurse and one field worker each, visited the contacts identified though the HDSS. Up to three attempts (visits) were made within 90 days of IC initial visit in order to recruit identified contacts. After providing informed consent, contacts were systematically screened following a 3-step approach: 1) HIV status verification: All participants with unknown or negative HIV status were offered HIV testing; 2) TB symptom screening (cough, hemoptysis, fever, weight loss and night sweats); and 3) sputum collection for microbiological testing for HIV negative participants who reported TB symptoms and HIV positive participants irrespective of symptoms (Fig 1). If participants were not able to provide spontaneous sputum, field sputum inductions were performed using portable nebulizers. Symptomatic patients were followed at the study facilities located in the Manhiça District Hospital. Clinical diagnosis was established by clinicians as per national guidelines (symptoms screening and X-ray interpretation). If clinical or lab-confirmed TB diagnosis was made, participants were referred to NTP facilities for treatment initiation. In case of pediatric contacts less than 12 years of age, additional TB symptoms included failure to thrive or reduced playfulness. If TB symptoms were present, attempts were made to obtain induced sputum, gastric aspirates, or, if available at the time of visit, urine of stool samples, who were later processed for Xpert Ultra.

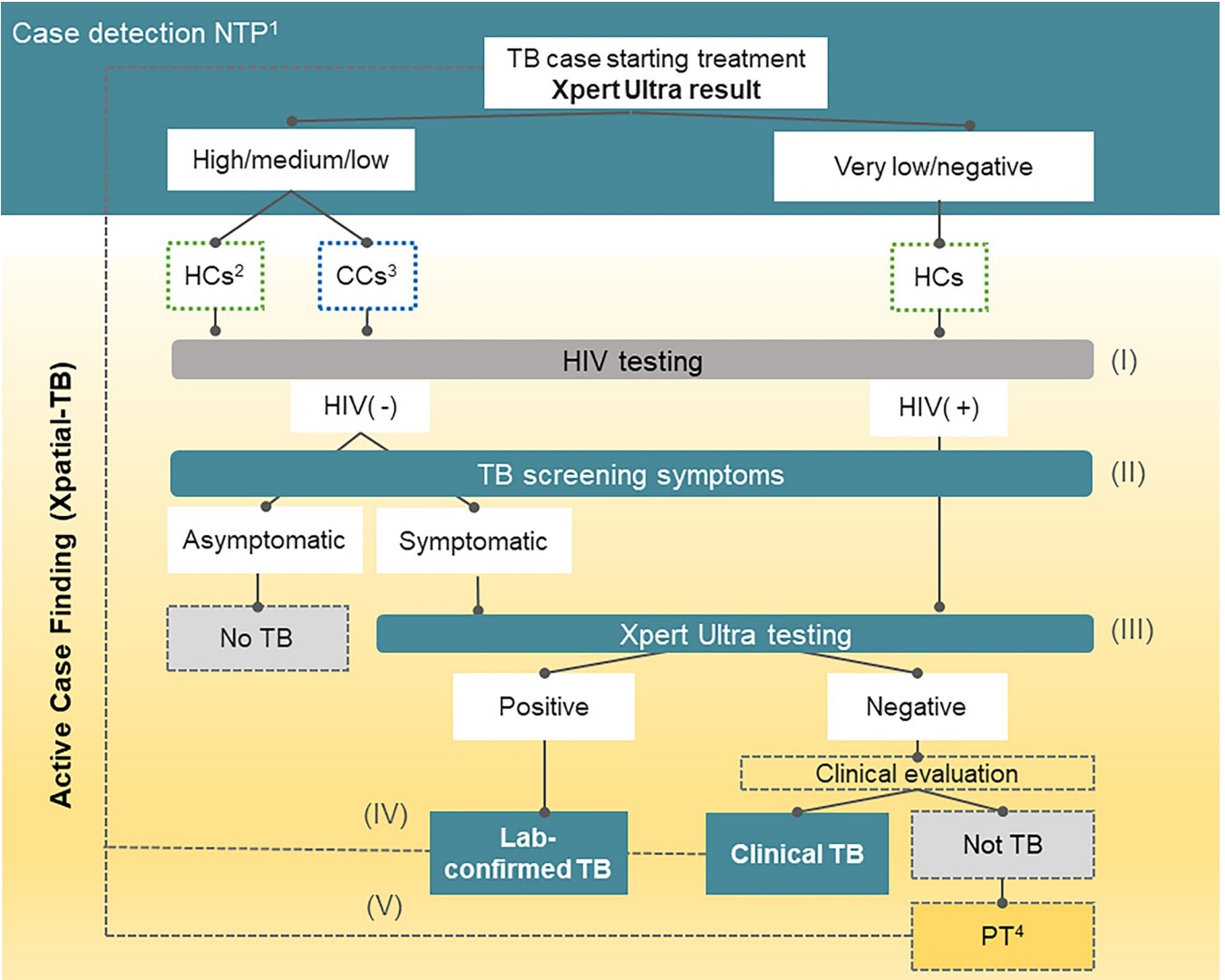

**Fig 1. Xpatial algorithm.** [1] NTP: National TB Programme; [2]HCs: household contacts; [3]CCs: close community contacts; [4]PT: Preventive therapy. The Active Case Finding Intervention had 5 steps: I) HIV testing; (II) Screening of TB symptoms; (III) Ultra testing for those accomplishing criteria (HIV positive regardless symptoms or symptomatic contacts); (IV) identified TB cases were referred to the NTP to started on treatment; (V) children < 5 years old were referred to started preventive therapy.

A more detailed description of the contact ratio calculations, diagnostic samples collection and laboratory methods can be found in the S1 Text Xpatial-TB procedures details.

## Data analysis

We assessed differences in the distribution of TB cases (detected by PCF or through our ACF strategy, in both HCs and CCs) by age, sex, demographic distribution, HIV status, bacteriological confirmation, Xpert Ultra results, TB type and X-ray results. Continuous variables were compared through means t-test and categorical variables through chi-square test.

The effect of the intervention was evaluated by: i) assessing the TB care cascade and describing process indicators (number of participants screened; number of participants enrolled, number or participants who provided sample, and number of patients diagnosed); ii) calculating numbers needed to screen (NNS) to find one case of TB as indicator of undetected TB and

efforts required to diagnose one case among different groups and epidemiological situations (NNS = 1/prevalence [22]); iii) conducting interrupted time series analysis with and without control (CITS/ITS) of aggregated quarterly TB notification cases [23]. Statistical analysis was performed using R version 3.5.2 (R Foundation for Statistical Computing, Vienna, Austria).

**Interrupted time-series (ITS) models.**   The ITS design is a quasi-experimental approach to evaluate health interventions introduced at a population level over a defined point in time [24]. On this basis, we aimed to assess the effect of the intervention through changes in TB notification rates (NR) across time.

Two periods were set: the pre-intervention (2015–2017), and the intervention period (2018). The pre-intervention period was used as a first counterfactual to make 2018 predictions in the absence of the intervention (before and after comparison). The outcomes of interest were: NR of TB and NR of laboratory-confirmed cases. We assumed no seasonality in TB notifications. We evaluated both, ITS-1: the introduction of the Xpert Ultra as the initial diagnostic test in the entire district, and ITS-2: the complete intervention (the use of Xpert-Ultra and ACF activities) by merging previous information with cases derived from ACF. Further information on ITS models can be found in the S2 Text. Interrupted time-series for single and multiple-group comparison.

**Control interrupted Time-series (CITS) model.**   Although the nature of the ITS design minimizes random within-group temporal changes, it cannot exclude other confounders, such as events around the time of the intervention that might have affected NRs [25]. Therefore, we included a control area as a second counterfactual to introduce a pooled control series. We chose notification data of 2 similar rural districts in Maputo province (Namaacha and Matutuine, Fig A in S1 Text), that do not limit with Manhiça district, in order to avoid the transfer of cases. Those areas applied smear microscopy as laboratory test for TB confirmation, and did not participate in other ACF projects in place or implemented the use of Xpert Ultra during 2018.

Population counts were obtained from the HDSS for Manhiça district. Census data were extracted from national projections for the control area.

We employed segmented linear regression to model the trend of the outcome over time [26]

The following equation was applied for the complete model: $Y_t = \beta_0 + \beta_1 T_t + \beta_2 X_t + \beta_3 X_t T_t + \beta_4 Z + \beta_5 Z T_t + \beta_6 Z X_t + \beta_6 Z X_t T_t$

Where $Y_t$ is the outcome at given point, T is time variable, X is a dummy variable representing the intervention (preintervention period as 0; intervention period as 1); and Z represents the intervention cohort (control area = 0; intervention area = 1). Model coefficients (β) were used to estimate the impact of the intervention by measuring outcome levels, trends and step/slope changes depending. Further details can be found in the S2 Text. Interrupted time-series for single and multiple-group comparison. Autocorrelation was tested by Breusch-Godfrey general test. Estimates were reported with standard errors and p-values as measures of precision. The final model was used to estimate a counterfactual of the notified cases in Manhiça district if the intervention had not occurred and difference with final cases reported. Further information on the CITS model can be found in S2 Text.

## Ethics considerations

The study was approved by the National Bioethics Committee for Health of Mozambique (CNBS, Ref:369/CNBS/17) and the Internal Bioethics Committee of CISM. All methods were performed in accordance with the relevant guidelines and regulations. All study participants (IC, HC and CC) signed written informed consent after a verbal explanation and written

information about the study was provided. For participants under 18 years of age, a written informed consent was obtained from their relatives, parents or guardians.

## Results

### Process indicators (Fig 2)

From January to December 2018, a total of 1010 new or relapse index TB cases, starting treatment, were included in the study (PCF cases). Following the Xpatial algorithm (Fig 1), an eligible population of 4394 HCs and 6373 CCs was identified by the HDSS; of those, 72.0% (3165/4394) and 74.2% (4730/6373) were respectively verbally screened for TB symptoms. This accounted for an average of 3.1 HCs (3165/1010) and 4.7 CCs (4730/1010) per index case. A total of 17.0% (1345/7895) were eligible for sampling (see Methods): 10.7% of all screened contacts (852/7812) who presented TB related symptoms (15.8% HCs vs 7.5% CCs, p< 0.001) and PLHIV without symptoms, which accounted for 6.2% (493/7812) of all contacts screened. We obtained sputum from 52.9% of them (712/1,345). Around 1.1% of all contacts screened (89/7,895) refused HIV testing.

As a result of the ACF strategy, 89 contacts were diagnosed with pulmonary TB, most of them symptomatic (79/89, 88.8%), but also 10 (11.2%) asymptomatic PLHIV (see methods). All of them started treatment. There was an overall microbiological confirmation of 7.8% (40/510) among symptomatic contacts who provided sample, and 3.4% (7/202) among asymptomatic PLHIV.

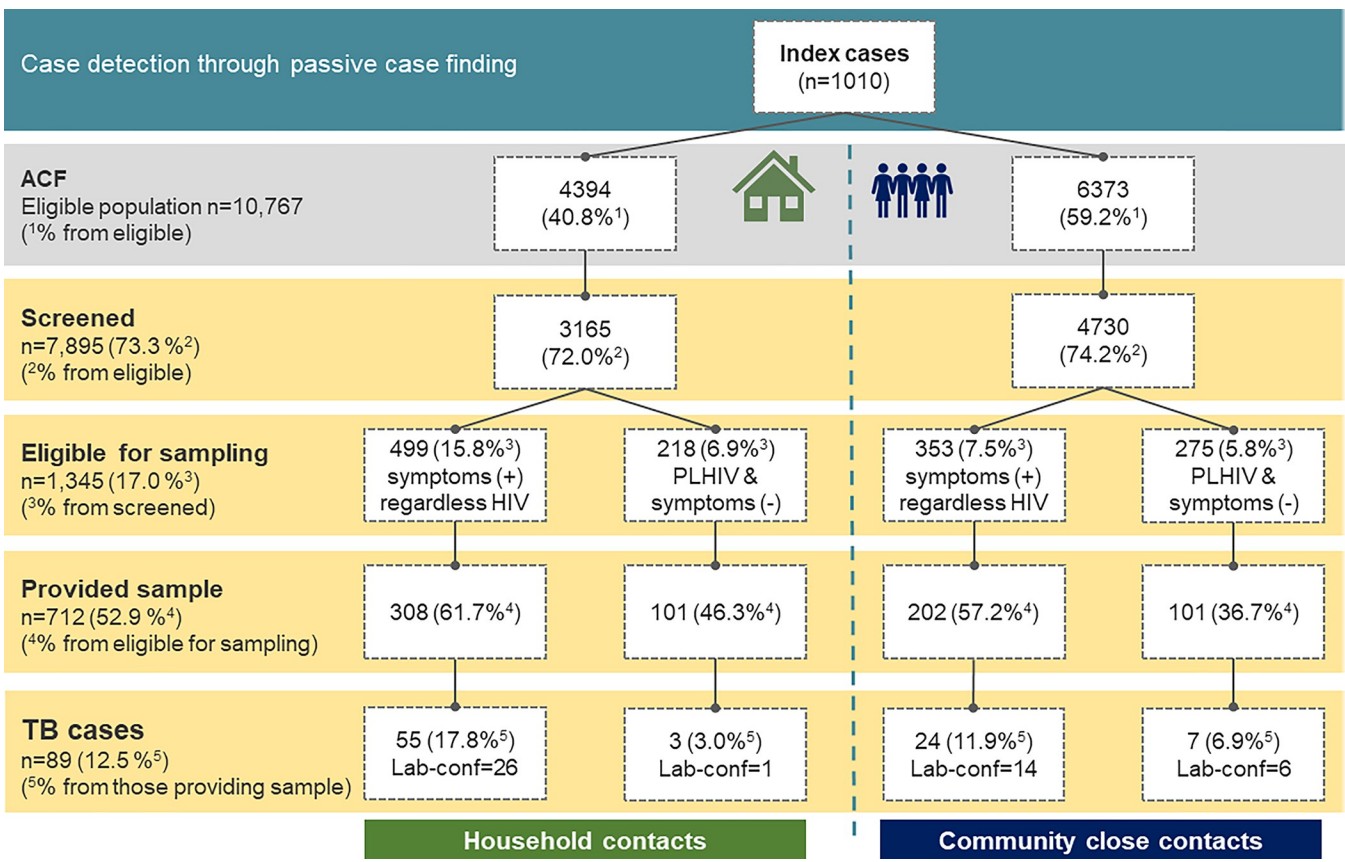

**Fig 2. Process indicators.** PCF: Passive Case Finding; ACF: Active Case Finding; Microb conf: microbiologically confirmed.

**Table 1. Number needed to screen for different study populations.**

|  | HCs[1] | CCs[2] | total |
|---|---|---|---|
| NNS[3] overall | 54.5 | 152.6 | 88.7 |
| NNS[4] lab-confirmed | 117.2 | 236.5 | 168.0 |
| NNS[5] symptomatic | 57.5 | 197.1 | 99.9 |
| NNS[6] asymptomatic PLHIV | 1055.0 | 675.7 | 789.5 |
| NNS[7] paediatric | 89.6 | 1102.0 | 190.8 |
| NNS[8] women | 113.0 | 236.5 | 164.5 |
| NNS[9] men | 105.5 | 430.0 | 192.6 |

1) HCs: Household contacts; 2) CCs: close community contacts; 3) NNS: number needed to screen to find a TB cases in the overall study population; 4) number needed to screen to find a laboratory—confirmed cases; 5) NNS to find a symptomatic patient; 6) number needed to screen to find an asymptomatic coinfected TB/HIV patient; 7) NNS to find a patient < 12 years old; 8) NNS to find a women with TB; 9) NNS to find a men with TB

The overall NNS to find a TB case was 88.7 cases, 54.6 for HCs and 152.6 for CCs. This indicator widely varied among specific groups (Table 1). It was necessary to screen 168 individuals to find 1 lab-confirmed case, 99.9 to find a symptomatic case, and 190.8 to find a paediatric case. Overall, NNSs were lower for HCs, except for asymptomatic PLHIV. Absolute numbers to calculate NNS are displayed in the S1 Table.

## Characteristics of cohorts

**Sociodemographic characteristics of index cases (ICs) (Table 2).** In 2018, 1020 incident TB cases were reported by the NTP. Individual information of those identified passively was available in 1010 patients (signed informed consent); the median age was 39.5 [IQR 30.1–52.4], 53.7% (543/1010) were male, and PLHIV accounted for 63.5% of the total of ICs (641/1010). Pulmonary disease was present in 92.7% (936/1010) of them. Overall, 37.1% of cases (375/1010) were microbiologically confirmed, 62.1% (233/375) of them being PLHIV. This translated into 36.4% (233/641) of bacteriological confirmation within this specific population, similar to the proportion of HIV-negative confirmed patients (38.6%, 142/367). Of those bacteriologically confirmed by Ultra, 58.5% (209/357) fell into the high or medium categories and 13.7% (49/357) into the 'trace'-call. Paediatric cases (children under 12 years of age) accounted for 6.1%. At the end of the study period, 10.1% (102/1010) of TB cases died. Just 3.1% (11/357) showed resistance to rifampicin by Ultra testing. Further details on ICs characteristics are shown in Table 1.

**Sociodemographic characteristics of screened population (S2 Table).** From those 1010 ICs, 7895 identified contacts were screened, and 55.7% (4398/7895) of them were women. The median age was 15.5 [8.0;32.4], showcasing different age distribution between HCs and CCs (32.0 IQR [9,45;54.1] vs 45.5 [32.9;54.3], p-value = 0.053). PLHIV accounted for 8.9% (701/7895) of evaluated contacts. This percentage was higher in HCs than in CCs (10.9% vs 7.5% p<0.001). More than one third of screened contacts (35.1%) belonged to Manhiça village health area.

**Sociodemographic characteristics of TB cases identified by the Xpatial intervention (Table 2).** From the total of 89 TB cases identified, 53.9% (48/89) were women. The median age was 40.2 [IQR:12.3; 54.3]. The age of cases diagnosed from HCs was significantly different from CCs (32.0 IQR [9.4;54.1] vs 45.5 [32.9;54.3], p-value = 0.05). Pediatric cases were found predominantly in household contacts (31.0% HCs vs 6.4% CCs); 25/27 children <15 and all children < 5 years old were found in household screening. The prevalence of HIV coinfection in this cohort was 49.4% (44/89), and 51.0% (24/47) among confirmed patients, being higher for HCs than CCs (59.3% vs 35.0%). Most TB cases had no previous history of TB (77/89, 86.5%).

**Table 2. Characteristics of index cases and TB cases diagnosed through the Xpatial strategy (overall and stratified by type of contact).**

| | ICs (PCF) [1] n = 1010 | p-value (PCF *vs* ACF) | Xpatial Strategy (ACF) [2] | | | p-value (HC vs CC) |
|---|---|---|---|---|---|---|
| | | | HCs [3] (n = 58) | CCs [4] (n = 31) | Total ACF (n = 89) | |
| | n (%) [5] | | n (%)[5] | n (%)[5] | n (%)[5] | |
| **Sex** | | 0.184 | | | | 0.214 |
| Women | 467 (46.3) | | 28 (48.3) | 20 (64.5) | 48 (53.9) | |
| Men | 543 (53.7) | | 30 (51.7) | 11 (35.5) | 41 (46.1) | |
| **Median Age [IQR][6]** | 39.5 [30.1;52.4] | 0.164 | 32.0 [9,45;54.1] | 45.5 [32.9;54.3] | 40.2 [12.3;54.3] | 0.053 |
| **Age group [7]** | | <0.001 | | | | |
| <5 | 28 (2.9) | | 8 (13.8) | 0 (0) | 8 (9.0) | |
| 5–15 | 41 (4.0) | | 17 (29.3) | 2 (6.4) | 19 (21.3) | |
| 15–35 | 319 (31.6) | | 5 (8.6) | 9 (29.0) | 14 (15.7) | |
| 35–55 | 402 (39.8) | | 15 (25.9) | 12 (38.7) | 27 (30.3) | |
| 55–75 | 184 (18.2) | | 8 (13.8) | 7 (22.6) | 15 (16.8) | |
| >75 | 35 (3.5) | | 5 (8.6) | 1 (3.2) | 6 (6.7) | |
| *Missing* | *1* | | | | | |
| **Paediatric TB cases** *(<12)* | 62 (6.1) | <0.001 | 18 (31.0) | 2(6.4) | 20 (22.5) | 0.008 |
| **Type of Tuberculosis** | | 1 | | | | 0.835 |
| New | 871 (86.2) | | 51 (87.9) | 26 (83.9) | 77 (86.5) | |
| Retreatment | 139 (13.8) | | 7 (12.1) | 5 (16.1) | 12 (13.5) | |
| **Tuberculosis location [8]** | | 0.003 | | | | 1 |
| Pulmonary | 936 (92.6) | | 58 (100) | 31 (100) | 89 (100) | |
| Extrapulmonary | 75 (7.4) | | 0 | 0 (0) | 0 (0) | |
| **Diagnosis** | | <0.001 | | | | 0.163 |
| Clinical diagnosis | 635 (62.9) | | 31 (53.4) | 11 (35.5) | 42 (47.2) | |
| Microbiological confirmation [9] | 375 (37.1) | | 27 (46.6) | 20 (64.5) | 47 (52.8) | |
| Prevalence HIV among confirmed | 233 (62.1) | 0.183 | 11 (40.7) | 13 (65.0) | 24 (51.0) | 0.177 |
| Confirmed HIV negative | 141 (37.6) | | 16 (59.3) | 7 (35.0) | 23 (49.0) | |
| *Xpert Ultra (n = 357)* | | <0.001 | | | n = 47 | 0.919 |
| *High* | 104 (29.1) | | 1 (3.7) | 1 (5.0) | 2 (4.2) | |
| *Medium* | 105 (29.4) | | 1 (3.7) | 2 (10.0) | 3 (6.4) | |
| *Low* | 53 (14.8) | | 4 (14.8) | 3 (15.0) | 7 (14.9)) | |
| *Very Low* | 46 (12.9) | | 7 (25.9) | 4 (20.0) | 11 (23.4) | |
| *Trace* | 49 (13.7) | | 14 (51.8) | 10 (50.0) | 24 (51.1) | |
| **HIV Status** | | 0.012 | | | | 0.158 |
| Positive | 641 (63.5) | | 25 (43.1) | 19 (61.3) | 44 (49.4) | |
| Negative | 367 (36.3) | | 33 (56.9) | 12 (38.7) | 45 (50.6) | |
| *Missing* | *2 (0.2)* | | | | | |
| **X-ray Results [10]** | | na | | | | na |
| Normal | 9 (0.9) | | | | | |
| Abnormalities not suggestive of TB | 24 (2.4) | | 1 (1.7) | 0 (0) | 1 (1.1) | |
| Abnormalities suggestive of TB | 555 (54.9) | | 28 (48.3) | 7 (22.6) | 35 (39.3) | |
| Abnormalities suggestive of past TB | 9 (0.9) | | - | - | - | |
| *Missing* | *413 (41)* | | 29 (50.0) | 24 (77.4) | 53 (59.6) | |
| **Mortality (deaths)** | 102 (10.1) | 0.09 | 1 | 3 (9.7) | 4 (4.5) | 0.119 |
| **Rif R (Xpert Ultra) [10]** | 11 (1.1) | | 2 (75) | 1 (25) | 3 (3.4) | |

1) ICs (PCF): Index cases (Passive Case Finding); 2) ACF: Active Case Finding; 3) HCs: Household contacts; 4) CCs: close community contacts; 5) column percentages; 6) IQR: Interquartile range; 7) 1 missing value; 8) 3 missing values 9) 18 cases confirmed by microscopy; 10) 79 not Xpert ultra result (n = 931)

More than half of cases was bacteriologically confirmed (52.8%, 47/89), and 74.5% (35/47) of them relied on Ultra´s lower categories (very low or trace). Three cases showed resistance to rifampicin (3.4%) and 4 patients died (4.5%) during follow-up.

Thirteen cases identified through PCF (9 women, 4 men) shared household with previously diagnosed PCF cases but were not detected by the ACF strategy. From those, only 9 had been registered as contacts; 2 were not reached, 4 were HCs screened and 1 CCs.

## Notification impact

As six ACF cases were diagnosed between January and March 2019 (window period), just 83 cases were used for the rate trend analysis.

**ITS-1: Evaluation of the implementation of Xpert Ultra as the frontline test in the district of Manhiça.**   As shown in Fig 3, between 2015–2017 (pre-intervention period) NR in Manhiça district showed a slight growing pattern up to 2017 (blue lines). Notifications then declined throughout 2017. This trend was reverted in 2018 (intervention period). Conversely, the proportion of laboratory confirmed cases was consistently low before the intervention (average of 23.5% in 2015–2017), showing a declining trend that was sharply reversed during the intervention (37.2%(375/1020)).

The ITS (interrupted temporal analysis for Manhiça district; Table 3, S1 Fig), estimated a pre-intervention positive trend among notified cases, with a small increase in quarterly NR ($\beta_1$ = 3.09 cases/100,000, SE = 1.24, p-value = 0.003). Although at the beginning of the intervention there was a reduction in the number of reported cases (-168/100,000, p-value = 0.1, following the downward trend which started in 2017, Fig 3), it was thereafter followed by an increasing NR ($\beta 3$ difference pre-post intervention = 9.54 /100,000)(Table 3).

The ITS confirmed a marked pre-intervention decline in the number of bacteriologically diagnosed cases (Fig B in S1 Fig); ($\beta_1$ = -2.18/100,000 per quarter, p-value = 0.007). This trend shifted during the intervention, with a positive difference ($\beta_3$) of 3.5 lab-confirmed cases /100,000 per quarter (p-value = 0.36)

**ITS-2: Evaluation of the implementation of the Xpatial intervention (Ultra + ACF).** The ACF intervention further increased by 8.2% all forms of TB cases detected in 2018 (83 ACF/1010 PCF) (Fig 3). The final proportion of bacteriologically confirmed TB cases during the intervention period was 38.3% (422/1103). ITS model parameters estimated a difference of 5.33 cases bacteriologically confirmed/100,000 population between the pre-intervention and the intervention period (Table 3, Fig 4, and Fig B in S1 Fig).

**CITS: Evaluation of the implementation of the Xpatial-TB intervention (Ultra + ACF) and comparison with the control area (Table 4 and Fig 4).**   For the CITS, the regression model identifies no overall significant-level or trend change between the control and the intervention area during the intervention period. The initial down step in number of cases reported occurred in both districts, as well as the recovery, although the decrease in NR was higher in the control area (difference step change $\beta_6$ of -50.68).

On the contrary, a significant difference in pre-intervention trends for microbiologically NR between areas was confirmed (-2.18 Manhiça vs -0.06 South Maputo, p-value = 0.012). This trend was positively reverted during the intervention period in Manhiça district (Fig 4), although the model did not show statistical significance.

## Discussion

To our knowledge, this was one of the largest ACF study conducted in household and community contacts in sub-Saharan Africa using Xpert Ultra as the frontline screening test, and which included both, spatial and microbiological parameters to select the target population. In

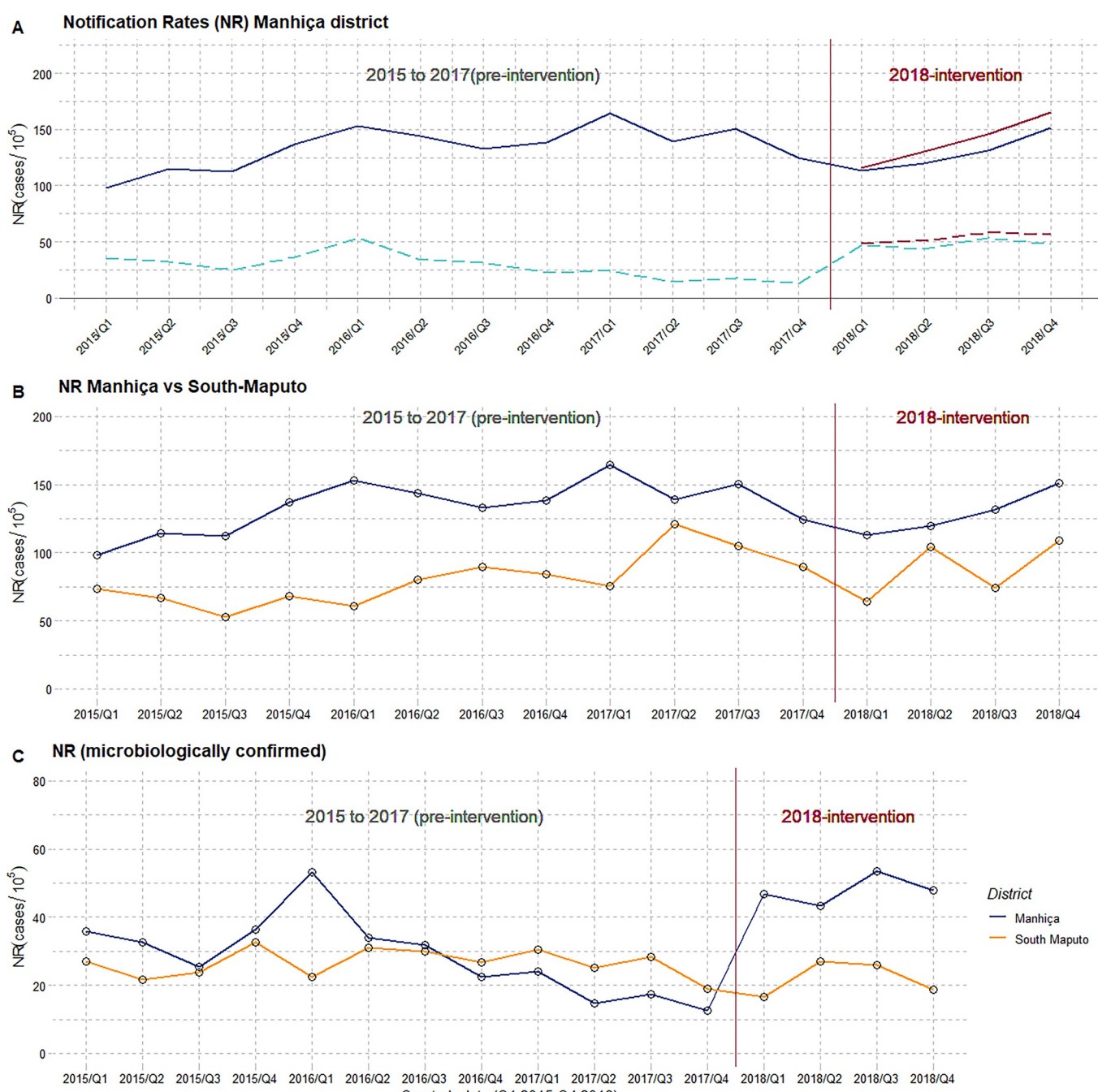

**Fig 3. A)** TB notification rates in Manhiça district pre-intervention/during intervention. TB cases excluding those derived from the ACF intervention are displayed in dark-blue. Additional ACF-cases are displayed in red. Bacteriologically confirmed cases are drawn in light-blue dash lines. **B)** TB Notification rates in Manhiça (blue) and the control area (Namaacha and Matutuine, yellow) **C)** Notification rates of microbiologically confirmed cases in Manhiça and the control area.

addition, we assessed the impact of using Xpert Ultra under programmatic conditions. The study shows: a) an overall increase in the number of lab confirmed cases and in the notification rate of microbiologically confirmed TB in the Manhiça district during the study period; b) a higher than expected proportion of paediatric cases diagnosed through the ACF strategy; c) a

**Table 3. Comparative of ITS model parameters for the total of cases (first column) and for bacteriologically confirmed cases (second column), based on quarterly case notification rates.** [#1] ITS-1: Evaluation of the introduction of Xpert Ultra as initial diagnostic test in Manhiça district; [#2] ITS-2: Evaluation of the Xpatial-TB intervention (Xpert Ultra + ACF activities).

| ITS-1 [#1] | NR Ultra[1] | SE[2] | p-value | NR Ultra (B+)[4] | SE | p-value |
|---|---|---|---|---|---|---|
| Trend prior intervention ($\beta_1$) | 3.09 | 1.24 | 0.03 | -2.18 | 0.67 | 0.007* |
| Change in level ($\beta_2$) | -168.28 | 96.56 | 0.11 | 13.81 | 52.71 | 0.80 |
| Difference between preintervention and intervention ($\beta_3$) | 9.54 | 6.72 | 0.18 | 3.50 | 3.37 | 0.36 |
| ITS-2 [#2] | NR Xpatial[3] | SE | p-value | NR Xpatial (B+) | SE | p-value |
| Trend prior intervention ($\beta_1$) | 3.09 | 1.24 | 0.03* | -2.18 | 0.67 | 0.007* |
| Change in level ($\beta_2$) | -211.62 | 95.90 | 0.05* | -34.79 | 51.63 | 0.51 |
| Difference between preintervention and intervention ($\beta_3$) | 13.26 | 6.68 | 0.07* | 5.33 | 3.59 | 0.16 |

[1]NR Ultra: notification rates evaluating the introduction of Xpert Ultra as the frontline test for diagnosis

[2]SE: Standard Error

[3]NR Xpatial: notification rates evaluating the complete intervention (Xpert Ultra+ACF)

[4]B+: bacteriologically confirmed. Autocorrelation of residuals by Breusch-Godfrey p-value = 0.54.

higher proportion of microbiologically-confirmation among TB cases diagnosed through ACF activities (compared to those diagnosed in PCF); and d) an increased number of Ultra positive results with low semiquantitative Xpert results among ACF cases (compared to PCF cases).

Undoubtedly, the integration of Xpert Ultra on programmatic case detection had a positive impact on the notification rate of laboratory confirmed cases and reversed the pre-intervention downward trend in the Manhiça district. Compared to the pre-intervention period, the proportion of lab-confirmed cases changed from 23.5 to 37.2% (excluding cases arising from ACF). The use of a more sensitive test for TB diagnosis is likely be the main reason for the

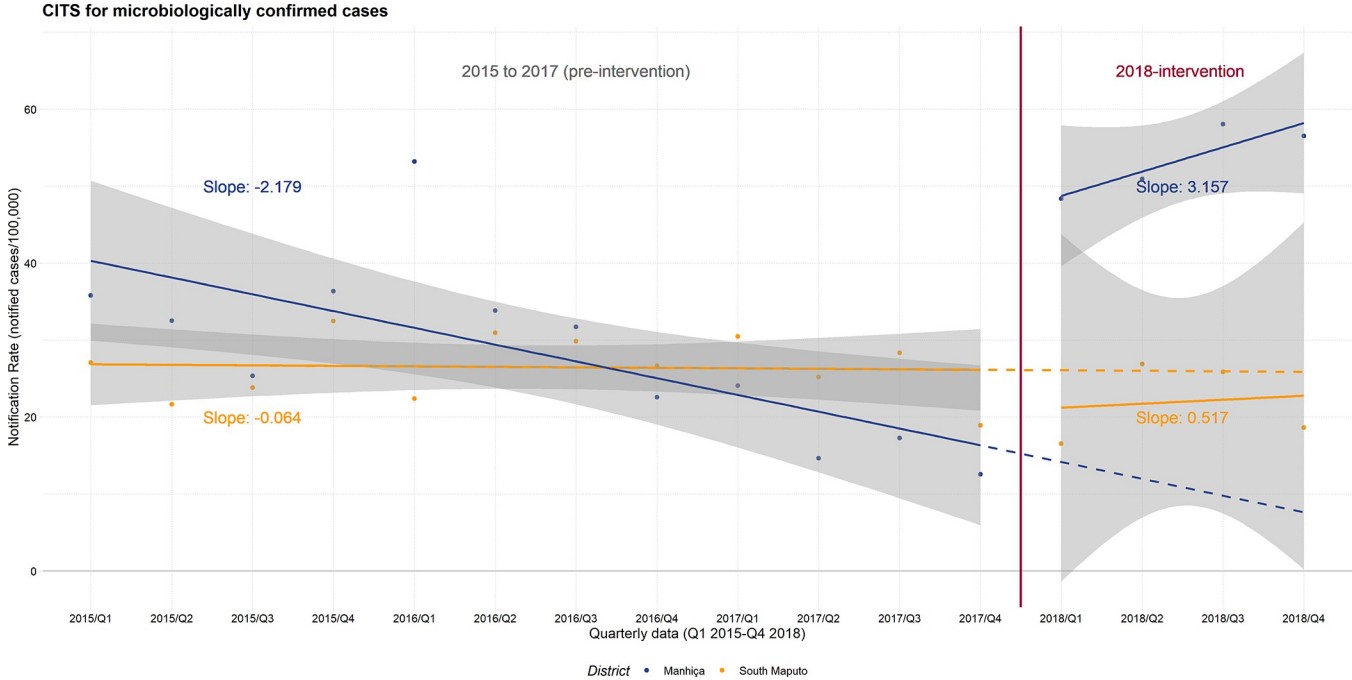

**Fig 4. Controlled interrupted series analysis (CITS) for bacteriologically confirmed cases.** Source of data: notification records from the National TB Programme. Control area: Districts of Namaacha and Matutuine, South Maputo; intervention area: Manhiça district. The pre-intervention period shows 3-years trend for reported confirmed cases. Dashed lines represent modelled trend (counterfactual estimates).

**Table 4. Comparative of CITS model parameters based on quarterly notification rates for all cases (first column) and just bacteriologically confirmed TB cases (second column), evaluating the Xpatial-TB intervention in comparison with the control area.**

|  | NR[1] Xpatial | SE[2] | p-value | NR Xpatial (B+)[3] | SE | p-value |
|---|---|---|---|---|---|---|
| Difference between preintervention and intervention trend, control ($\beta_3$) | 6.49 | 6.79 | 0.71 | 0.58 | 3.01 | 0.19 |
| Difference in baseline intercepts ($\beta_4$) | 58.46 | 12.99 | <0.001* | 15.58 | 5.76 | 0.012* |
| Difference in pre-intervention trends ($\beta_5$) | -0.77 | 1.76 | 0.66 | -2.11 | 0.78 | 0.012* |
| Difference in intervention step change ($\beta_6$) | -94.01 | 137.4 | 0.5 | -22.37 | 60.24 | 0.71 |
| Difference in intervention trends ($\beta_7$) | 6.77 | 9.57 | 0.49 | 4.75 | 4.19 | 0.27 |

[1]NR notification rates

[2]SE: Standard Error

[3]B+: bacteriologically confirmed Autocorrelation of residuals by Breusch-Godfrey p-value = 0.57

considerable difference in lab-confirmed cases in 2018. Indeed, 13.7% of notified TB cases reported on Ultra's new lowest category, "trace". If the former Xpert MTB/RIF assay had been used instead, it is likely that those cases would have not been confirmed (and a proportion of them would have been not diagnosed for TB).The initial debate on the significance of trace results for the Xpert Ultra cartridge seems to have achieved a certain consensus. The latest recommendations advocate against retesting in favour of a cautious assessment of the clinical presentation and individuals' characteristics [27].

The overall proportion of microbiological confirmation was also considerably lower in the PCF cohort (37.1%), and aligned with the annual country report (39%), when compared to the ACF cohort (52.8%) [28]. The low confirmation rate is been repeatedly questioned by the WHO since the national TB prevalence survey [10]. If a widely-applied screening found more lab-confirmed cases, we may suspect a large number of hidden and missed patients in the district. Moreover, the lower bacillary burden found in samples from ACF activities would likely mean that we found patients with less severe case disease, meeting one of the central objectives of screening activities: early detection of cases to interrupt the transmission chain [5, 29].

Although the comparison with the control area did not show significant difference in TB notification (overall or lab-confirmed cases) using CITS analyses, there is a clear shift in the trend and amount of confirmed cases for Manhiça district that did not happen in the control area. It is likely that the few data points available from the NTP reports (quarterly data, four points), might have played a role in the absence of statistically significant results. Similar to other ACF intervention studies, we used a linear regression to model the data. However, we could not include other covariates that might have affected the diagnosis of cases and explain irregular reporting over time (stock-outs of reagents, health coverage or retention in care data). A higher number of control districts, and the assessment of post intervention trends, would have helped to formally elucidate whether our intervention had a positive impact in CITS analyses.

Prior to the study, ACF activities were inconsistent in the district and relied on the availability of community-health workers. The implementation of a massive screening strategy, targeting pre-defined high-risk populations increased the overall number of detected cases by 8.9%, and the number of laboratory-confirmed cases by 15.7%. Evidence on the effectiveness of community-based ACF activities vary widely among studies [8], and determining whether our strategy was affordable and cost-effective would depend on economic evaluations [22]. Nevertheless, process indicators and differences found among screened/PCF/ACF cohorts may provide information on likely sources of detection gaps in our district and populations who would benefit the most from the intervention.

Based on NNSs (common indicator to evaluate ACF activities [4]), the yield of HCs screenings was considerably higher than for CCs. NNSs were higher for CCs than for HCs in almost all cases (i.e. overall NNS to find a TB case was 54.6 HCs vs 152.6 CCs). Those differences were even higher among paediatric cases (NNS HCs 89.9 in HCs vs NNS 1102 in CCs). Since TB disease among children results from persistent close contact with an infectious IC, this finding would support the benefit of intensive household screening and confirm the low paediatric TB case detection in our setting, as previously described [30]. Reported data on childhood TB is an important indicator when evaluating NTPs. For the same year, Mozambique reported that 13% of cases were under 15 years old, far from the 6% identified in the district. This reinforces the need to take urgent actions to increase TB detection in this vulnerable population, such as integrated TB care interventions of the NTP and maternal and child health programmes [5, 31].

The prevalence of HIV/TB cases identified through the NTP was over 60%, matching previous reports [14]. These figures are consistently higher than those reported for the country as a whole (36%) and justified (at the time when the study was designed) the strategy of universal testing for HIV individuals regardless of symptoms. Surprisingly, the number of TB/HIV cases found was lower than expected for this specific population (overall NNS of 789.5) and far from that reported in similar studies [32]. Likewise, the prevalence of HIV among screened contacts was also lower than previously seen, and specific analysis are being conducted to better understand the HIV burden in several populations of the district. Of note, the specific NNS for finding females and males was similar among HCs, but differed significantly in CCs (NNS to find a man/woman among screened: 430 vs 226). Although the local population pyramid (ratio F/M 1.2) [20], would skew the sex distribution of screened individuals and ACF cases (54% of females), this did not happened for HCs. Therefore, further gendered research should be conducted to gain better understanding on the role of CCs screening in finding more women.

Given the high NNS for CCs and the low yield reached on community cases, we wondered whether TB cases identified through CCs were actually cases epidemiologically linked to the IC (or, alternatively, co-prevalent cases). Although the district has some populated areas, most inhabitants live in rural, open and spacious compounds. Prolonged daily interaction with infected people may take place in other places, such as public transport, bars or workplaces [33]. Molecular epidemiology studies should shed more light on the nature of the link between IC and TB cases found among contacts [34–36]. In addition, although the study might have played a key role in Manhiça's uptake and integration of the new Ultra testing to routine care, we lack consistent data to evaluate post-intervention years. Furthermore, the number of cases identified through the ACF strategy was lower than initially expected. It can be argued that the small window screening period used (3 months), might be too short to find recent contacts [35], or the existence of incident cases misdiagnosed as clinical cases.

Questions such as the optimal window period for screening, whether there is a need for time-separated screening visits, the role of HIV status in TB transmission and the influence of sex determinants on the impact of ACF activities need to be answered to refine ACF intervention designs. Lastly, TB control might no longer be based on the historical dichotomy, either active or metabolically inactive latent infection [37]. If subclinical cases can transmit the infection in the community, traditional contact-tracing interventions based on TB compatible symptoms- should be expanded to testing asymptomatic, HIV negative contacts.

## Conclusion

Through the implementation of a combined strategy, the scaling-up of the Ultra test for programmatic activities and the implementation of an innovative ACF strategy, more bacteriologically confirmed TB cases were reported in the district of Manhiça. However, our CITS

analysis could not confirm the impact of the strategy on either TB case notification or bacteriologically confirmed cases, potentially due to limitations in the analytic approach and the limited data on confounding factors. Paediatric population benefited the most from the ACF strategy and HCs screening was confirmed as an effective strategy to find microbiological confirmed cases in early stages of the disease.

## Supporting information

**S1 Text. Xpatial TB procedures details.** Table A. Screening radii calculations, Fig A. Maps.
(DOCX)

**S2 Text. Interrupted time-series for single and multiple-group comparison.**
(DOCX)

**S1 Table. Absolute numbers to calculate NNS (number needed to screen).**
(DOCX)

**S2 Table. Characteristics of screened household and community contact.**
(DOCX)

**S1 Fig. Visualization of interrupted time series analysis for Manhiça district.** Fig A. Visualization of interrupted time series analysis for Manhiça district (notification of total number of cases), Fig B. Visualization of interrupted time series analysis for Manhiça district (notification of microbiologically confirmed cases).
(DOCX)

**S1 Data. Xpatial tb master.**
(CSV)

## Acknowledgments

The authors are grateful to all of the research participants who generously gave their time and effort to this project. We would also like to acknowledge the contributions of the National TB Programme and healthcare workers of the Ministry of Health of Mozambique. We thank Dr Oriol Ramis, Dr Joan Cayla and Dr Joan Pau Millet for his assistance during the project and Dr J. Gandasegui for his contribution with Figs 1 and 2, and his inputs.

## Author Contributions

**Conceptualization:** Laura Oliveras, Alberto L. García-Basteiro.

**Data curation:** Edson Mambuque, Gustavo Tembe, Paulo Philimone.

**Formal analysis:** Belén Saavedra, Laura de la Torre-Pérez, Matthew Rudd, Paulo Philimone.

**Funding acquisition:** Alberto L. García-Basteiro.

**Investigation:** Belén Saavedra, Dinis Nguenha, Laura de la Torre-Pérez, Edson Mambuque, Gustavo Tembe, Laura Oliveras, Paulo Philimone, Juan Ignacio Garcia, Neide Gomes, Shilzia Munguambe, Helio Chiconela, Milton Nhanommbe, Santiago Izco, Sozinho Acacio, Alberto L. García-Basteiro.

**Methodology:** Belén Saavedra, Paulo Philimone, Alberto L. García-Basteiro.

**Project administration:** Belén Saavedra.

**Resources:** Alberto L. García-Basteiro.

**Supervision:** Dinis Nguenha, Juan Ignacio Garcia, Milton Nhanommbe, Sozinho Acacio, Alberto L. García-Basteiro.

**Validation:** Belén Saavedra, Laura de la Torre-Pérez.

**Visualization:** Belén Saavedra, Laura de la Torre-Pérez.

**Writing – original draft:** Belén Saavedra, Laura de la Torre-Pérez.

**Writing – review & editing:** Belén Saavedra, Dinis Nguenha, Laura de la Torre-Pérez, Edson Mambuque, Gustavo Tembe, Laura Oliveras, Matthew Rudd, Paulo Philimone, Benedita Jose, Juan Ignacio Garcia, Neide Gomes, Shilzia Munguambe, Helio Chiconela, Milton Nhanommbe, Santiago Izco, Sozinho Acacio, Alberto L. García-Basteiro.

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
