## [Decision Letter · Decision Letter 0]

17 Jan 2023

PGPH-D-22-01781

Improving tuberculosis case detection through contact risk stratification by Xpert MTB/RIF Ultra and spatial parameters: Evaluation of an innovative active case finding strategy in Mozambique (Xpatial-TB)

Dear Dr. García-Basteiro,

Thank you for submitting your manuscript to PLOS Global Public Health. After careful consideration, we feel that it has merit but does not fully meet PLOS Global Public Health’s publication criteria as it currently stands. Therefore, we invite you to submit a revised version of the manuscript that addresses the points raised during the review process.

We look forward to receiving your revised manuscript.

Kind regards,

Raquel Muñiz-Salazar, Ph.D.

Academic Editor

Journal Requirements:

1. Please amend your detailed online Financial Disclosure statement. This is published with the article. It must therefore be completed in full sentences and contain the exact wording you wish to be published.

a) Please clarify all sources of financial support for your study. List the grants, grant numbers, and organizations that funded your study, including funding received from your institution. Please note that suppliers of material support, including research materials, should be recognized in the Acknowledgements section rather than in the Financial Disclosure.

b) State the initials, alongside each funding source, of each author to receive each grant. For example: "This work was supported by the National Institutes of Health (####### to AM; ###### to CJ) and the National Science Foundation (###### to AM)."

c) State what role the funders took in the study. If the funders had no role in your study, please state: “The funders had no role in study design, data collection and analysis, decision to publish, or preparation of the manuscript.”

2. Please update your online Competing Interests statement. If you have no competing interests to declare, please state: “The authors have declared that no competing interests exist.”

3. In the online submission form, you indicated that "An anonymized dataset will be available upon request to the corresponding author or the internal scientific committee of Manhiça at cci@manhica.net". All PLOS journals now require all data underlying the findings described in their manuscript to be freely available to other researchers, either 1. In a public repository, 2. Within the manuscript itself, or 3. Uploaded as supplementary information.

4. We have noticed that you have uploaded Supporting Information files, but you have not included a list of legends. Please add a full list of legends for your Supporting Information files after the references list.

Additional Editor Comments (if provided):

The manuscript is attractive and well-designed but lacks an analytical approach. The manuscript is attractive and well-designed but lacks an analytical approach. This relevant study combines Xpert Ultra semi-quantitative and spatial parameters on tuberculosis case notifications in a high TB-HIV burden district of Southern Mozambique.

The two reviewers have made important observations on the manuscript, which must be addressed for the article could be accepted.

Reviewers' comments:

Reviewer's Responses to Questions

**Comments to the Author**

1. Does this manuscript meet PLOS Global Public Health’s publication criteria? Is the manuscript technically sound, and do the data support the conclusions? The manuscript must describe methodologically and ethically rigorous research with conclusions that are appropriately drawn based on the data presented.

Reviewer #1: Yes

Reviewer #2: Yes

2. Has the statistical analysis been performed appropriately and rigorously?

Reviewer #1: No

Reviewer #2: Yes

3. Have the authors made all data underlying the findings in their manuscript fully available (please refer to the Data Availability Statement at the start of the manuscript PDF file)?

Reviewer #1: No

Reviewer #2: Yes

4. Is the manuscript presented in an intelligible fashion and written in standard English?

Reviewer #1: Yes

Reviewer #2: Yes

5. Review Comments to the Author

Reviewer #1: Overall, the authors submitted a well-written manuscript describing the programmatic outcomes of a novel active case finding method for TB. Major comments and methodological concerns include:

- Conclusion in abstract states that the intervention led to increase in TB case notifications. This seems like an overly strong statement of causality given that the differences between control areas could not be established. Therefore, the increase in TB case notifications could be due to secular trends and possible detection of cases that would have also been detected via passive case detection. While I believe ACF likely had a beneficial effect, I think the conclusion should be more nuanced.

- I think the manuscript could be strengthened by greater attention to the possible impact of over-diagnosis of TB in the study. It is striking that <40% of index cases were microbiologically confirmed TB. The relatively low yield of TB among contacts could also reflect that a significant portion of the index cases did not have TB. On the other hand, nearly half of contacts diagnosed with TB were not bacteriologically confirmed, which could have led to over-estimation of the yield of ACF. A few suggestions:

o In Methods, please describe additional details about the TB diagnosis process in Mozambique with a focus on three districts. For example, if Xpert MTB/RIF was not being used in the 2 control districts, were sputum microscopy primarily used? What are the clinical diagnostic criteria for laboratory unconfirmed TB? Likewise, for Manhica, Xpert Xpert/MTB Ultra was used as the “front line” test. What was the process of diagnosis among Xpert negative cases?

o Please consider performing sensitivity analysis limiting the data to lab confirmed TB. Do the findings and overall conclusion differ?

o Please add a description of the effect of possible inclusion of non-TB index cases in the limitations.

- Given the objective of the study and data collected, I question whether ITS is the appropriate analytical method for this study. It seems that a simple comparison of the notification rate before vs. after and between intervention and control districts would be sufficient. I find the ITS findings overly difficult to interpret. I suggest removing the ITS analysis from the manuscript. However, it the authors do keep ITS analysis, please consider the following:

o Please explain your hypothesis of your ITS analysis. Would you expect a step change or slope increase? Why?

o Tables 3 and 4 include NR Ultra and NR Xpatial coefficient estimates. It is unclear from the methods how you can distinguish the effect of Ultra vs. the combined intervention since both Ultra and ACF were implemented at the same time. Please clarify in the Methods and on the table.

o Figure 4 is very difficult to understand. Please add details to the figure legends and add labels to the figure as needed to clarify what is being presented. Also, please confirm that the colors are correctly labelled. As it stand, it appears that there was a big step increase in case notification among control districts post-intervention, but a decrease in Manhica.

Minor comments and feedback include:

- Line 72 The citation field needs to be updated.

- Line 93 “quality or” should be corrected to “quality of”

- Line 105. I believe the authors meant “exposure” not “exposition”

- Line 118. The study is described as a “prospective controlled intervention study”. It is unclear from this definition, how the study is controlled. I suggest adding details about the methods of control here or remove “controlled” from this description.

- Figures 1 and 2 are very clear and helpful for understanding the study. Thank you for including them.

- Line 143 “Once and IC” should be “Once an IC”

- Line 143 and throughout paper: Since HDSS data were used to identify neighbors for screening, I believe it’s incorrect to call them “contacts”. In general, “contacts” are used to refer to people who had physical contact with the index case.

- Please justify why epidemiological contact history was not taken (or if it was, why it was not used to guide contact investigation).

- Line 178. Please describe the specific variables of the TB care cascade and process indicators that were included in the outcome analysis.

- Line 209. Continuum should be continuous.

- The use of existing community census data to guide ACF is novel. Please add details on the outcome of that effort. For example, what was the average number of HH contacts and neighborhood residents screened per index?

- Table 1. If NNS women (8) and NNS men (9) represent the total men and women needed to screen to diagnose one women or men, they do not seem meaningful to me. Please justify why this was calculated or remove.

- In Results, Please consider placing Characteristics of cohort before the description of contact screening results.

- Results. Please show the yield by Xpert Ultra bacillary burden categories.

- Results. Please show yield by type of sample (e.g. sputum, urine, stool)

- In Discussion, please place NNS in context with prior household contact studies.

- Limitations. Please include the lack of epidemiological contact information available to guide ACF.

Reviewer #2: Comments to Authors:

Introduction

1. Line 67, cite latest TB prevalence rates from latest WHO report

2. Line 90, Use full form of any terminology at its first use.

3. Line 91, what do you mean by detection gap?

Methods

4. Xpatial Strategy: Fig.1: provide the footnote.

What does numbers 1,2 and 3 in superscript forms within the fig 1 indicate?

While dividing the Xpert Ultra results in two categories, what does low category in both bifurcations means? Explain

What do you mean by clinical TB? The resolution of figure is very low.

Results

5. Table 1: What was the purpose behind screening gender-wise NNS (NNS women, NNS men)? Why NNS women among women and men were selected top screen? The purpose of superscript number is not mentioned in footnote.

6. Sociodemographic characteristics of index cases: Line 264-267, “In 2018 1020 incident TB cases ……… women and men”, statement is not clear.

7. Line 267-269, “PLHIV …… pulmonary disease”, make the statement clearer.

8. Numbers are not matching in table 2 and text for microbiological confirmation and confirmed HIV negative and positive and so on.

9. Line 290, Pediatric cases: children <15 or <12? Explain.

10. Table 3 make separate columns for pre and post intervention for NR ultra and NR Xpatial

11. Line 346-349, p values and β1 and β3 values seems confusing. Please make a clear table 3

12. Line 367, number are not matching with the table 4. Make table 4 clear for different districts and pre post intervention periods.

13. Figure 4 make resolution clearer and explain the graph or slope how it is showing a positive trend.

14. There is no molecular evidence whether these HCc and CCs belong to the true index cases they were tracing, or contacts got infection from some other sources if they had migrated or travelled during this time.

Discussion

15. Your objective is to see the trend in improvement of TB case detection using ACF strategy not to see the index cases trend in pre- and intervention period. What’s the purpose of looking at this trend in pre-intervention period?

16. “Indeed, 13.7% of notified TB cases reported on Ultra's new lowest category, "trace". If the former Xpert assay had been used instead, it is likely that those cases would have not been confirmed (and a proportion of them would have been not diagnosed for TB). The initial debate on the significance of trace results for the Xpert MTB/RIF Ultra cartridge seems to have achieved a certain consensus. The latest recommendations advocate against retesting in favor of a cautious assessment of the clinical presentation and individuals’ characteristics “. Of course, new technology will have some improvement tin case detection rather than old. What’s the reason behind bringing this discussion here?

Typographical and grammatical errors

6. PLOS authors have the option to publish the peer review history of their article (what does this mean?). If published, this will include your full peer review and any attached files.

**Do you want your identity to be public for this peer review?** For information about this choice, including consent withdrawal, please see our Privacy Policy.

Reviewer #1: **Yes: **Sanghyuk Shin

Reviewer #2: **Yes: **Ravi Prakash

---

## [Decision Letter · Decision Letter 1]

22 Jun 2023

PGPH-D-22-01781R1

Improving tuberculosis case detection through contact risk stratification by Xpert MTB/RIF Ultra and spatial parameters: Evaluation of an innovative active case finding strategy in Mozambique (Xpatial-TB)

Dear Dr. García-Basteiro,

Thank you for submitting your manuscript to PLOS Global Public Health. After careful consideration, we feel that it has merit but does not fully meet PLOS Global Public Health’s publication criteria as it currently stands. Therefore, we invite you to submit a revised version of the manuscript that addresses the points raised during the review process.

We look forward to receiving your revised manuscript.

Kind regards,

Raquel Muñiz-Salazar, Ph.D.

Academic Editor

Journal Requirements:

Additional Editor Comments (if provided):

The authors have taken into account most of the feedback from the previous review. However, one major issue is that the conclusion statement in the abstract lacks evidence from the data presented. According to the abstract, there was a rise in laboratory-confirmed TB cases in the intervention district as compared to the control area, as per the CITS model. However, Table 4 does not reveal any significant variation in the step change or intervention trends of the two areas. Rephrase this statement to ensure it corresponds with the outcomes without changing its meaning, adding any new information, new sentences or paragraphs, and removing any important information from the text. Please use the same formality style for the rewrite as the one used in the original text.

Furthermore, some minor comments require attention before the acceptance of the manuscript.

Reviewers' comments:

Reviewer's Responses to Questions

**Comments to the Author**

1. If the authors have adequately addressed your comments raised in a previous round of review and you feel that this manuscript is now acceptable for publication, you may indicate that here to bypass the “Comments to the Author” section, enter your conflict of interest statement in the “Confidential to Editor” section, and submit your "Accept" recommendation.

Reviewer #1: All comments have been addressed

Reviewer #2: All comments have been addressed

2. Does this manuscript meet PLOS Global Public Health’s publication criteria? Is the manuscript technically sound, and do the data support the conclusions? The manuscript must describe methodologically and ethically rigorous research with conclusions that are appropriately drawn based on the data presented.

Reviewer #1: Partly

Reviewer #2: Yes

3. Has the statistical analysis been performed appropriately and rigorously?

Reviewer #1: Yes

Reviewer #2: Yes

4. Have the authors made all data underlying the findings in their manuscript fully available (please refer to the Data Availability Statement at the start of the manuscript PDF file)?

Reviewer #1: Yes

Reviewer #2: Yes

5. Is the manuscript presented in an intelligible fashion and written in standard English?

Reviewer #1: Yes

Reviewer #2: (No Response)

6. Review Comments to the Author

Reviewer #1: The authors addressed the majority of my comments from the earlier review. My major concern is that the concluding statement in the abstract is not supported by the data presented. The abstract states “The CITS model showed an increase of laboratory confirmed TB cases in the intervention district, compared to the control area”. However, Table 4 shows no statistically significant difference in step change or intervention trends between the two areas. Please rephrase this statement so that it is aligned with the results.

Minor comments include:

- Of the 10 asymptomatic PLHIV diagnosed with TB, 7 were bacteriologically confirmed (Line 230). Please explain the justification for TB diagnosis of the 3 asymptomatic PLHIV with no bacteriologic confirmation.

- In line 258, please specify that 1010 refers to index cases and 7895 refers to people who were identified as contacts.

- Please note that the use of p-values in Table 2 is not consistent with best statistical practice (see Wasserstein et al. below). I recommend removing the p-values from Table 2. If the authors insist on using p-values, they should be consistently reported for all variables. At present, p-values are presented for some variables but not all in Table 2 (e.g. Prevalence of HIV, age group).

- 1.

Wasserstein RL, Lazar NA. The ASA’s Statement on p-Values: Context, Process, and Purpose. The American Statistician. 2016 Apr 2;70(2):129–33.

- Line 414 states that the NNS shows higher effectiveness of HC screening. However, NNS is a measure of efficiency, not effectiveness. I recommend changing the term “effectiveness” to “efficiency” or “yield”.

- Line 434. I suggest changing “NNS to find a man/woman” to “NNS to find a TB case among men/women”.

- It would be helpful to know if the sex differences in NNS among community contacts were observed in prior TB ACF studies. Suggest adding context to this finding based on existing literature, if any.

- It seems that the uploaded csv file contains dates of birth, which could be used to identify study participants. Suggest removing this column from the shared data file.

Reviewer #2: (No Response)

7. PLOS authors have the option to publish the peer review history of their article (what does this mean?). If published, this will include your full peer review and any attached files.

**Do you want your identity to be public for this peer review?** For information about this choice, including consent withdrawal, please see our Privacy Policy.

Reviewer #1: No

Reviewer #2: **Yes: **Ravi Prakash

---

## [Decision Letter · Decision Letter 2]

15 Nov 2023

PGPH-D-22-01781R2

Improving tuberculosis case detection through contact risk stratification by Xpert MTB/RIF Ultra and spatial parameters: Evaluation of an innovative active case finding strategy in Mozambique (Xpatial-TB)

Dear Dr. García-Basteiro,

Thank you for submitting your manuscript to PLOS Global Public Health. After careful consideration, we feel that it has merit but does not fully meet PLOS Global Public Health’s publication criteria as it currently stands. Therefore, we invite you to submit a revised version of the manuscript that addresses the points raised during the review process.

We look forward to receiving your revised manuscript.

Kind regards,

Cesar Ugarte-Gil, MD, MSc, PhD

Academic Editor

Journal Requirements:

Additional Editor Comments (if provided):

First, apologies for the delayed response to the authors. The majority of the comments and concerns were solved by the authors, but a reviewer provides some fair comments that I consider important to address and/or explain as potential limitations before to accept it for publication. Considering the delay in the process, as a Editor I will check them as soon the authors submit them

Reviewers' comments:

Reviewer's Responses to Questions

**Comments to the Author**

1. If the authors have adequately addressed your comments raised in a previous round of review and you feel that this manuscript is now acceptable for publication, you may indicate that here to bypass the “Comments to the Author” section, enter your conflict of interest statement in the “Confidential to Editor” section, and submit your "Accept" recommendation.

Reviewer #3: (No Response)

2. Does this manuscript meet PLOS Global Public Health’s publication criteria? Is the manuscript technically sound, and do the data support the conclusions? The manuscript must describe methodologically and ethically rigorous research with conclusions that are appropriately drawn based on the data presented.

Reviewer #3: No

3. Has the statistical analysis been performed appropriately and rigorously?

Reviewer #3: No

4. Have the authors made all data underlying the findings in their manuscript fully available (please refer to the Data Availability Statement at the start of the manuscript PDF file)?

Reviewer #3: Yes

5. Is the manuscript presented in an intelligible fashion and written in standard English?

Reviewer #3: No

6. Review Comments to the Author

Reviewer #3: Synopsis

This study aimed to assess the impact of implementing Xpert Ultra as the primary test for diagnosing tuberculosis (TB) combined with an innovative active-case finding (ACF) strategy based on Xpert Ultra semi-quantitative results and spatial parameters in a semi-rural area of southern Mozambique. The ACF component was conducted in 2018. Household contacts (HCs), and community contacts (CCs) of index cases (IC) were recruited according to semi-quantitative Xpert Ultra results and variable radii determined by the neighbourhood population density of the IC. Interrupted time series linear regression was used to compare pre- and post-intervention notification rates, and laboratory-confirmed notification rates both within the intervention district and to selected non-intervention control areas. While the study utilised a novel ACF strategy and addresses an important question, substantial methodological issues unfortunately make it unsuitable for publication.

Major issues:

1. TB case definition and diagnosis: Only 375/1010 (37.1%) of the index cases (ICs) had laboratory-confirmed TB. Furthermore 568(89%) of the 635 clinically diagnosed ICs had actually a negative Xpert Ultra result (from data file provided, numbers not included in the manuscript). 49/357 (13.7%) of the laboratory-confirmed IC cases were classified as laboratory-confirmed TB on the basis of trace positive Xpert results without taking previous TB history into account. As a result of the above, it is very uncertain whether the IC, or the ACF, actually had TB, particularly given the very high proportion of ICs which had a negative Xpert test.

2. Analytical methods: The authors note a definitively non-linear trend in notification rates (NR) in pre-intervention period (2015-2017) within the intervention district (lines 295-297) with NR increasing between 2015 and 2016 but declining in 2017. The use of linear regression ITS/CITS is therefore inappropriate as it averages the pre-intervention trend across the period and is likely the main driver of the large level decrease (B2 coefficients, table 3) in the in ITS-1 and ITS-2 models. While non-linear methods may somewhat address the issue, there is another underlying problem in that the data for the 2015 quarters were derived from aggregated reports from the NTP whereas the 2016-2017 (and ?2018 data) were obtained from individual level data raising potential concerns about different data processing and therefore comparability of the newer data with the 2015 data.

3. Selection of control areas: The study combined data from 2 non-invention areas to form a control for the CITS analyses. It is unclear how or why these areas were selected and more information is required. (As an aside it appears from the supplementary material that a different control area “Magude” which is in a very different area may have originally been considered.) The substantial differences in the pre-intervention intercepts and pre-intervention slopes (B4 and B5 coefficients respectively, table 4) raise concerns about the conclusions drawn from the CITS models. Alternative control areas which result in B4 and B5 coefficients with higher p-values should be considered if at all possible.

Minor issues:

1. Include map of the intervention and control areas.

2. The paper requires substantial editing for grammatical and language issues. A few examples are given below (by no means an exhaustive list):

a. Abstract line 39: Change “rolling-up” to “rolling out”

b. Abstract line 43: rephrase “on Xpert Ultra ´s result of the IC “with “on the Xpert Ultra result of the IC”

c. Abstract line 43/44: rephrase “population density of the area where the IC lived in” to “population density of the area in which the IC lived”

d. Abstract line 45: Edit “asked for providing” to “asked to provide”

e. Line 110: “approximately 204,953” – suggest round to nearest 1000 to avoid spurious accuracy.

f. Line 260: “being the age distribution different between HCs and CCs” is unclear

g. Line 287: Write “BK” in full as abbreviation not used elsewhere

3. The proportion of ACF requiring sputum induction with portable nebuliser would assist in interpreting the feasibility of the ACF intervention in settings without portable nebulisers.

4. Line 194 in methods: ”Autocorrelation was tested by Cumby–Huizinga” but results of this test are not presented.

5. Line 254: “Just 1.1% (11/1010) showed resistance 254 to rifampicin by Ultra testing” but denominator includes both those with negative Xperts and without Xpert tests.

6. Table S1: Screening radius for medium density areas not provided

7. Supplement S2.1 and S2.2: it appears that the coding for the intervention period dummy variable Xt: could be incorrect (dummy variable; pre-intervention (before 2018)=1/intervention(2018)=0)” as the B coefficients will be the value in the pre-intervention period where X==1 and 0 in the post intervention period where X==0.

8. Reference issues (again not a complete list)

a. Line 114: reference 21 is incorrect as this reference is for South Africa and does not relate to the HDSS

b. References (7) and (13); (14) and (32); and (30) and (35) are all duplicate sets of references.

7. PLOS authors have the option to publish the peer review history of their article (what does this mean?). If published, this will include your full peer review and any attached files.

**Do you want your identity to be public for this peer review?** For information about this choice, including consent withdrawal, please see our Privacy Policy.

Reviewer #3: **Yes: **Harry Moultrie

---

## [Editor Report · Decision Letter 3]

18 Dec 2023

Improving tuberculosis case detection through contact risk stratification by Xpert MTB/RIF Ultra and spatial parameters: Evaluation of an innovative active case finding strategy in Mozambique (Xpatial-TB)

PGPH-D-22-01781R3

Dear Dr. García-Basteiro,

We are pleased to inform you that your manuscript 'Improving tuberculosis case detection through contact risk stratification by Xpert MTB/RIF Ultra and spatial parameters: Evaluation of an innovative active case finding strategy in Mozambique (Xpatial-TB)' has been provisionally accepted for publication in PLOS Global Public Health.

Best regards,

Cesar Ugarte-Gil, MD, MSc, PhD

Academic Editor